# Inverse design of 3D reconfigurable curvilinear modular origami structures using geometric and topological reconstructions

Kai Xiao [1], Zihe Liang[1], Bihui Zou[1], Xiang Zhou [2] ✉ & Jaehyung Ju [1] ✉

The recent development of modular origami structures has ushered in an era for active metamaterials with multiple degrees of freedom (multi-DOF). Notably, no systematic inverse design approach for 3D curvilinear modular origami structures has been reported. Moreover, very few modular origami topologies have been studied to design active metamaterials with multi-DOF. Herein, we develop an inverse design method for constructing 3D reconfigurable architected structures − we synthesize modular origami structures whose unit cells can be volumetrically mapped into a prescribed 3D curvilinear shape followed by volumetric shrinkage to construct modules. After modification of the tubular geometry, we search through all the possible geometric and topological combinations of the modular origami structures to attain the target mobility using a topological reconstruction of modules. Our inverse design using geometric and topological reconstructions can provide an effective solution to construct 3D curvilinear reconfigurable structures with multi-DOF. Our work opens a path toward 3D reconfigurable systems based on volumetric inverse design, such as 3D active metamaterials and 3D morphing devices for automotive, aerospace, and biomedical engineering applications.

With the advancement of conventional manufacturing in both additive[1] and subtractive[2] ways, the fabrication of complex 3D structures is being realized on nano[3–7], micro[8–12], meso[13–19], and large scales[20–22]. Researchers are even pushing the limit of complex 3D structural design to motion structures that can change their shapes from one state to another, tuning their physical properties in adapted and active ways to varying physical environments[23].

Many groups have employed origami and kirigami techniques, the ancient folding and cutting arts[24], given the potential of future intelligent reconfigurable structures. Starting from 2D flexible structures using a Miura-ori sheet[25], Waterbomb base[26], or Ron-Resch pattern[27], researchers have recently explored the design of 3D architected structures using modular origami. Stacking multiple origami sheets[28,29] and assembling foldable modules[15,16,30–33] are the currently available methods to construct 3D architected structures. Stacked Miura-ori[28] and Tachi−Miura polyhedron[30] are deployable with flat-foldability, with a single origami sheet serving as the building block. Assembled structures with folding modules using prismatic polyhedrons[15,16,33], tubular bellows[31,34,35], and kirigami[32] have demonstrated the potential of isotropic design using spatial symmetry while displaying single and multiple degrees of freedom (DOFs)[15,16,33].

All these methods require the construction of a building block (unit cell) with crease patterns and spatial periodicity (tessellation). This bottom-up design approach is convenient for controlling the kinematic and kinetic properties of the overall 3D architected structures with only a unit-cell design. However, this bottom-up design

[1]UM-SJTU Joint Institute, Shanghai Jiao Tong University, 800 Dongchuan Road, Shanghai, China. [2]School of Aeronautics and Astronautics, Shanghai Jiao Tong University, 800 Dongchuan Road, Shanghai, China. ✉e-mail: xiangzhou@sjtu.edu.cn; jaehyung.ju@sjtu.edu.cn

approach has a critical limitation when constructing practical engineering and artistic structures, the shapes of which are mostly 3D curvilinear (e.g., automotive and aerospace structures with various curvatures). Such structures require that the size and shape of the building block no longer be homogeneous in the design domain. Most rectangular or cubic engineering structures have curved edges for minimum stress concentration and safety. Because of the break in spatial periodicity at the boundary of curvilinear geometries, the kinematic and kinetic properties of the unit cell tend to differ from those of the tessellated structure, limiting the structural implementation of the architected origami materials.

The top-down approach to the design, often called 'inverse design,' of mechanical metamaterials has been explored to identify tessellated microstructures and target physical properties such as anisotropic stiffness. Topology optimization has been a typical approach to identify microstructures[36–38]. Very few studies have explored the inverse design of 3D architected structures[39]. The existing inverse-design methods can only be applied to 2D curvilinear surfaces with origami and kirigami[40–47] and not to volumetric 3D spatial curvilinear geometries and their reconfigurability.

In this work, we explore an inverse design for 3D reconfigurable architected origami materials. Without tessellating a constant building block, our method produces volumetric gradient cells mapped into complex curvilinear 3D geometries, followed by topological reconstruction of modules. We develop a top-down approach making unit cells map into a sphere, hyperboloid, cone, twisted cylinder, torus, and any combined curvilinear shapes with reconfigurability.

## Results

### Synthesis of nonperiodic modular origami

Inspired by the pioneering work on the space-filling tessellation to construct 3D modular origami with prismatic tubes[16], we implement the synthesis principle to 3D curvilinear shapes with nonperiodic tiling of irregular polyhedrons. Figure 1 presents our synthesis procedure for 3D curvilinear modular origami structures. A unit cell consisting of single or multiple polyhedrons is first selected, as shown in Fig. 1a. A nonperiodic tessellation into a prescribed curvilinear geometry is then applied, as illustrated in Fig. 1b, to build a template, such as that presented in Fig. 1c. While constructing the template, we implement a volumetric mapping of the unit cell using the optimum transport algorithm[48] for a target number of tessellations, e.g., $2 \times 2 \times 3$ in Fig. 1c, to obtain deformed irregular polyhedrons generated by the minimum energy for deformation. During the volumetric mapping, the polyhedrons in unit cells are involved in nonhomogeneous deformation.

Next, we spatially shrink the deformed polyhedrons in the template while implementing a scaling constraint, as illustrated in Fig. 1d:

$$S_{a_{1,p}} = S_{b_{1,p}} = S_{a_{2,p}} = S_{b_{2,p}}, \tag{1}$$

where the scaling ratios $S_{a_{i,p}} = \frac{|\mathbf{a}_{i,p} - \mathbf{A}_p|}{|\mathbf{o}_i - \mathbf{A}_p|}$ and $S_{b_{i,p}} = \frac{|\mathbf{b}_{i,p} - \mathbf{B}_p|}{|\mathbf{o}_i - \mathbf{B}_p|}$ $(i = 1, 2)$. $\mathbf{o}_i$ is the centroid of the deformed $i$-th polyhedrons in the template. $\mathbf{A}_p$ and $\mathbf{B}_p$ belong to the edge of the $p$-th face shared by two adjacent polyhedrons in the template. The scaling constraint in Eq. (1) forces the direction of connection on the $p$-th face to be parallel after shrinking, i.e., $\mathbf{a}_{1,p} - \mathbf{a}_{2,p} = \mathbf{b}_{1,p} - \mathbf{b}_{2,p}$, critical for building a connection with adjacent shrunken polyhedrons. Our work is different from previous work[16], where the connecting direction was normal to the regular polyhedrons; instead, we determine the connecting direction by bridging the centroids of the deformed polyhedrons in the template. We obtain the length of the connection $L_p$ between the adjacent shrunken polyhedrons:

$$L_p = S_{a_{1,p}} |\mathbf{o}_2 - \mathbf{o}_1|. \tag{2}$$

Next, we connect the shrunken polyhedrons by extruding prismatic tubes on each shared face inside the template, as shown in Fig. 1e. Equations (1) and (2) are applied to the generation of tubes to connect adjacent shrunken polyhedrons. We apply a different construction method for the tube on the exterior boundary surface of a shrunken polyhedron, which was at the boundary of the template before the volumetric shrinkage. The tubular length $d_b$ generated along the exterior boundary surface in Fig. 1d is determined by the distance between the template and the exterior surface of a shrunken polyhedron along its average normal direction. After the geometric reconstruction, we obtain a nonperiodic modular origami structure with a curvilinear boundary, as shown in Fig. 1e. Note that the connected tubes serve as structural components, whereas the shrunken polyhedrons function as porous holes. See the synthesis in Supplementary Movie 1.

We can build other complex nonperiodic modular origami structures using the geometric reconstruction with the volumetric mapping into an arbitrary target shape and volumetric shrinkage of deformed polyhedrons, as illustrated in Fig. 1f–j. Additional examples of the geometric reconstruction are provided in Supplementary Fig. 3 in the Supplementary Information (SI). Although we focus on nonperiodic modular origami structures in this work, our synthesis method of the spatial architected materials can also be applied to planar cases, as illustrated in Supplementary Fig. 1 of the SI, demonstrating that our design method is universal.

### Construction of reconfigurable structures

Although the geometric reconstruction generates arbitrary 3D curvilinear structures through volumetric mapping and volumetric shrinkage of unit cells, the constructed origami structures hardly produce reconfigurability. From a macroscopic perspective, the fully connected origami modules with fully extruded tubes produce spatial loops, constraining the motion of the modules[49]. Note that a module denotes a single origami unit constructed by the extruded tubes from the shrunken polyhedrons after geometric reconstruction, as illustrated in Fig. 2a. One can select rigid or flexible modules depending on their geometry and topology. From a microscopic perspective, the extruded prismatic tubes of irregular shrunken polyhedrons limit the range of motion of the whole assembly, as shown in Fig. 2 (see also Supplementary Fig. 4 of the SI).

To release the microscopic immobility, extruded tubes generated by the geometric reconstruction in Fig. 1e–j or Fig. 2a follow the geometric modification procedure:

$$\min \sum \left\| \mathbf{v}_i - \bar{\mathbf{v}}_i \right\| \tag{3}$$

$$s.t. \quad \mathbf{I}_t \cdot \mathbf{d}_t = \mathbf{0}, \quad \text{for } t = 1, 2, 3, \ldots, N_t \tag{4}$$

$$\mathbf{I}_t \cdot \mathbf{d}_t = \mathbf{0} \quad \text{for } t = 1, 2, 3, \ldots, N_t$$

$$\mathbf{J}_q \cdot \boldsymbol{\theta}_q = \mathbf{0}, \quad \text{for } q = 1, 2, 3, \ldots, N_q \tag{5}$$

$$\mathbf{p}_{i_j} \cdot \left( \mathbf{p}_{1_j} \times \mathbf{p}_{2_j} \right) = 0. \quad \text{for } j = 1, 2, 3, \ldots, N_j \tag{6}$$

The objective function in Eq. (3) minimizes the change of the nodal position $\mathbf{v}_i - \bar{\mathbf{v}}_i$ on the boundary of the initial architected materials in Fig. 2a, where $\mathbf{v}_i$ and $\bar{\mathbf{v}}_i$ are the adjusted and initial position vectors of the $i$-th node on the boundary, respectively. During the modification, we allow the initial modular origami

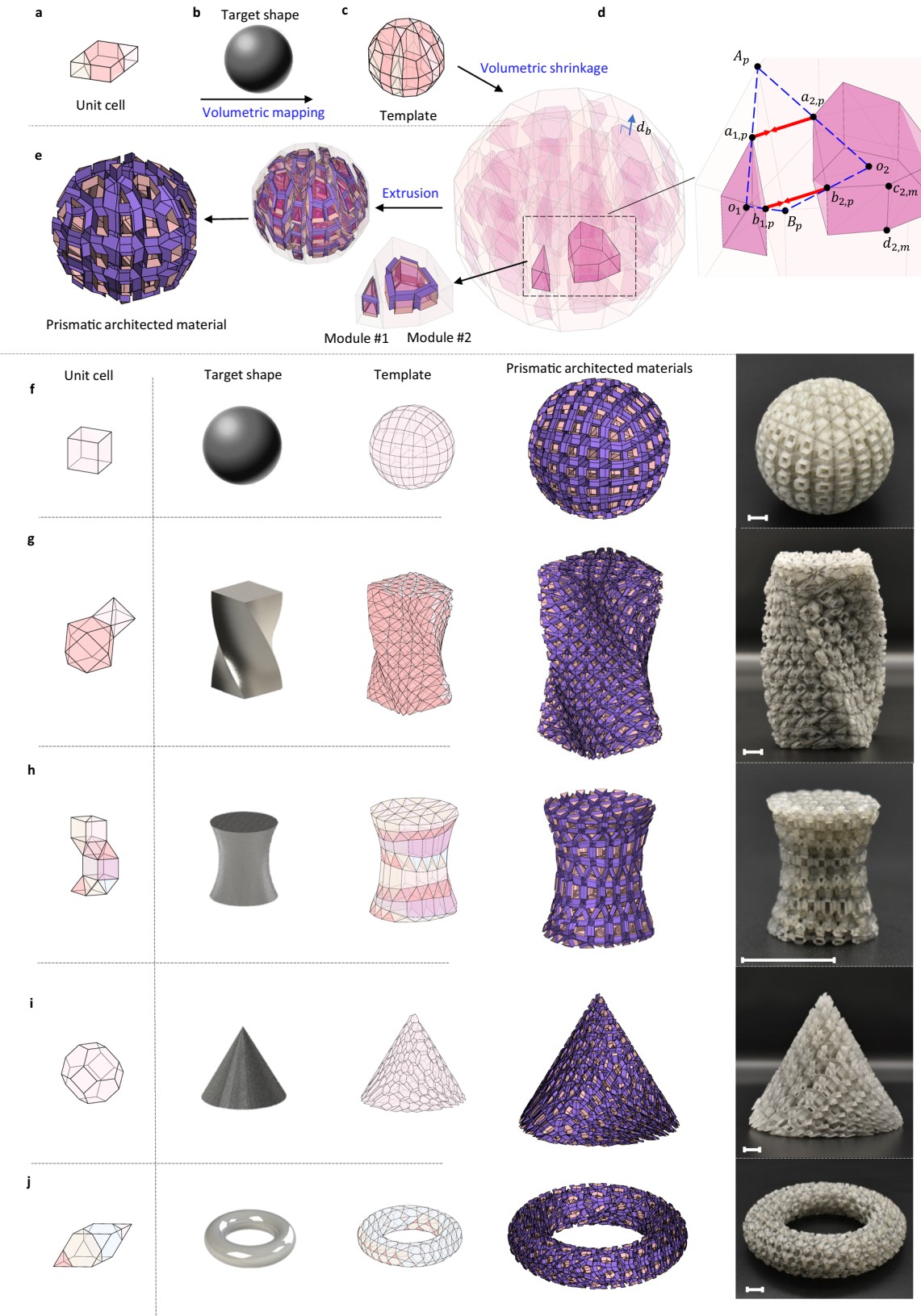

**Fig. 1 | Geometric reconstruction of nonperiodic modular origami structures.** **a** Unit cell with polyhedrons, **b** target shape, and **c** template with deformed polyhedrons. The details of the volumetric mapping from 'a' to 'c' are provided in Methods. **d** Volumetric shrinkage; the shrunken polyhedrons are highlighted in purple. **e** Extrusion of prismatic tubes to connect adjacent shrunken polyhedrons. **f**–**j** Other examples of the geometric reconstruction for synthesizing 3D architected materials with modular origami. The rightmost column shows the 3D-printed prototypes (see Supplementary Note 4 of the SI for fabrication details). Scale bar, 1 cm.

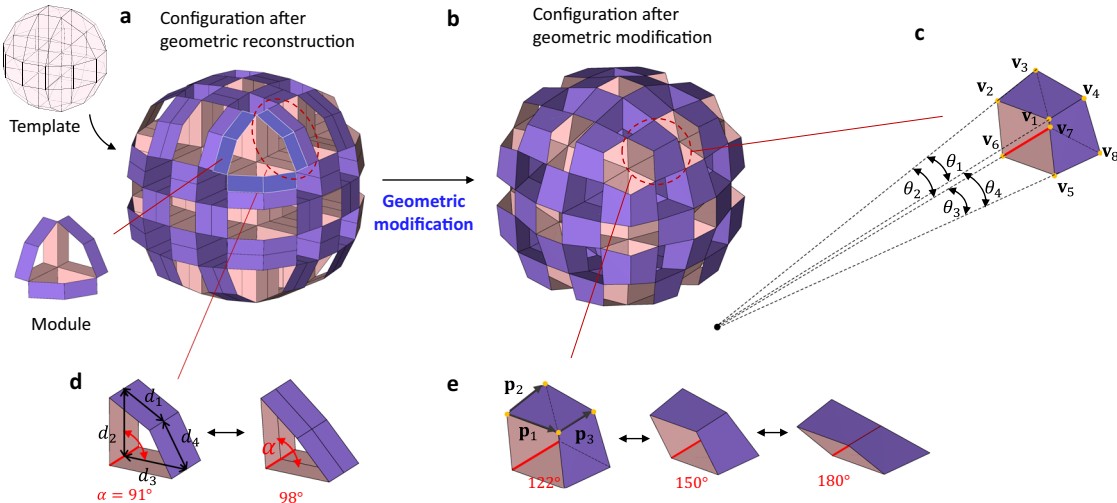

**Fig. 2 | Geometric modification. a** Initial configuration of modular origami structures of a sphere generated by volumetric mapping and volumetric shrinkage (3×3×3) of a cubic unit cell. Note that modules can be built by selective tubular extrusion of shrunken polyhedrons. **b** Configuration after geometric modification. Equations (4) and (5) produce the constraints for the rectangular tubes, i.e., $d_1 = d_2 = d_3 = d_4$ and $\theta_1 = \theta_2 = \theta_3 = \theta_4$. **c** A tube with intersecting hinges, where $\theta_1$ is the angle between the projected $\mathbf{v}_4 - \mathbf{v}_1$ and $\mathbf{v}_3 - \mathbf{v}_2$, and $\theta_2$ is the angle between

$\mathbf{v}_3 - \mathbf{v}_2$ and $\mathbf{v}_1 - \mathbf{v}_6$. $\theta_3$ and $\theta_4$ are similarly obtained. The extended lines of these four hinges ($\mathbf{v}_1 - \mathbf{v}_4, \mathbf{v}_2 - \mathbf{v}_3, \mathbf{v}_6 - \mathbf{v}_7, \mathbf{v}_5 - \mathbf{v}_8$) intersect at a point. **d** A tube with parallel hinges having limited foldability in the initial configuration. **e** A tube with intersecting hinges having flat-foldability in the modified configuration with a planar constraint of $\mathbf{p}_3 \bullet (\mathbf{p}_1 \times \mathbf{p}_2) = 0$, where vectors $\mathbf{p}_1$, $\mathbf{p}_2$, and $\mathbf{p}_3$ are on the edges of a face.

structures after the geometric reconstruction to have a combination of parallel and intersecting tubes whose numbers are $N_t$ and $N_q$, respectively. Figure 2c, d presents examples of intersecting and parallel tubes, respectively. Equation (4) implements the foldability of parallel tubes by maximizing the range of motion of the prismatic tubes with parallel hinges; $\mathbf{d}_t = \left[ d_{1_t}, d_{2_t}, d_{3_t}, \ldots, d_{f_t} \right]$, where $f$ is the number of faces of the $t$-th tube. $\mathbf{I}_t$ is a constraint matrix whose components are determined by the extruded polygon shape; see more details of the constraints in Supplementary Note 2 of the SI. Note that Eq. (4) is the flat-foldable constraint for the distance $d$ of the extruded polygons. Equation (5) adjusts the motion of the tubes whose hinges intersect to a point. Equation (5) constrains the angle $\theta$ between adjacent intersecting hinges; $\boldsymbol{\theta}_q = \left[ \theta_{1_q}, \theta_{2_q}, \theta_{3_q}, \ldots, \theta_{e_q} \right]$, where $e$ is the number of side walls of the $q$-th tube. $\mathbf{J}_q$ is a constraint matrix whose components are determined by the polygon shape of the side wall of the extruded tubes. Further details on the constraints are provided in Supplementary Note 2 of the SI. Equation (6) ensures that all $N_j$ faces remain planar. For a single face $j$, $\mathbf{p}_{i_j}$ is the vector attached to the $i$-th edge of the $j$-th face.

Figure 2 shows the geometric modification of an architected structure constructed using the geometric reconstruction. Unlike the initial architected structure in Fig. 2a, the modified configuration in Fig. 2b makes the prismatic tubes deformable to their maximum range of motion while maintaining the macroscopic sphere shape. Note that the prismatic tubes in the initial configuration have limited motion, as illustrated in Fig. 2d; however, the prismatic tubes of the modified configuration are flat-foldable, as illustrated in Fig. 2e. Notably, the geometric modification maximizes the sole foldability of individual prismatic tubes.

After the geometric modification, we search through all the possible geometric and topological combinations of each module, applying the graph theory in the assembly. A graph consists of vertices and edges representing modules and tubular connections, as shown in

Fig. 3a.3, a.8. To attain the target mobility $\bar{N}_{dof}$, we formulate a search algorithm of reconfigurable structures.

$$\min \left( N_{dof}(\mathbf{x}) - \bar{N}_{dof} \right)^2 \qquad (7)$$

$$s.t. \quad x_i \in D_i, \quad \text{for } i = 1, 2, 3, \ldots, N_x \qquad (8)$$

$$-C_i(\mathbf{x}) + 2 \leq 0, \qquad (9)$$

$$C_p(\mathbf{x}) - 1 = 0. \qquad (10)$$

where $x_i$ represents the $x_i$-th module and $D_i$ is a collection of all modules for the $i$-th shrunken polyhedron. $N_x$ is the total number of modules in the assembly. $N_{dof}(\mathbf{x})$ is the DOF for the combination $\mathbf{x} (= x_i \hat{\mathbf{e}}_i)$, where $\hat{\mathbf{e}}_i$ is a basis of vector $\mathbf{x}$. The constraints in Eqs. (9) and (10) control the structural integrity of the assembly, where $C_i(\mathbf{x})$ is the number of connections on the $i$-th node and $C_p(\mathbf{x})$ is the number of connected graph components. We solve this optimization problem using a genetic algorithm; see more details in Supplementary Note 2 of the SI.

Figure 3a shows the inverse design while emphasizing the topological reconstruction, e.g., constructing a reconfigurable structure generated by a sphere template comprising deformed irregular tetrahedra and octahedra. After the geometric reconstruction, the initial modular origami structure is modified using Eqs. (3)–(6) before the topological reconstruction.

Depending on the topologically reconstructed modules selected, e.g., Fig. 3a.5, a.6, we can construct assemblies whose graphs are fully connected with seventeen basic cycles ($n_{bc} = 17$, Fig. 3a.4) or partially connected with fewer basic cycles ($n_{bc} = 3$, Fig. 3a.9), providing reconfigurable structures with $n_{DOF} = 0$ or 10, respectively, as shown in Fig. 3a.2, a.7 (see also Supplementary Movie 2). As demonstrated by the numerical simulation and experiments in Fig. 3b, this new structure can change its shape from a sphere to other shapes via global motion at the hinges (see the transformation in Supplementary Movie 3).

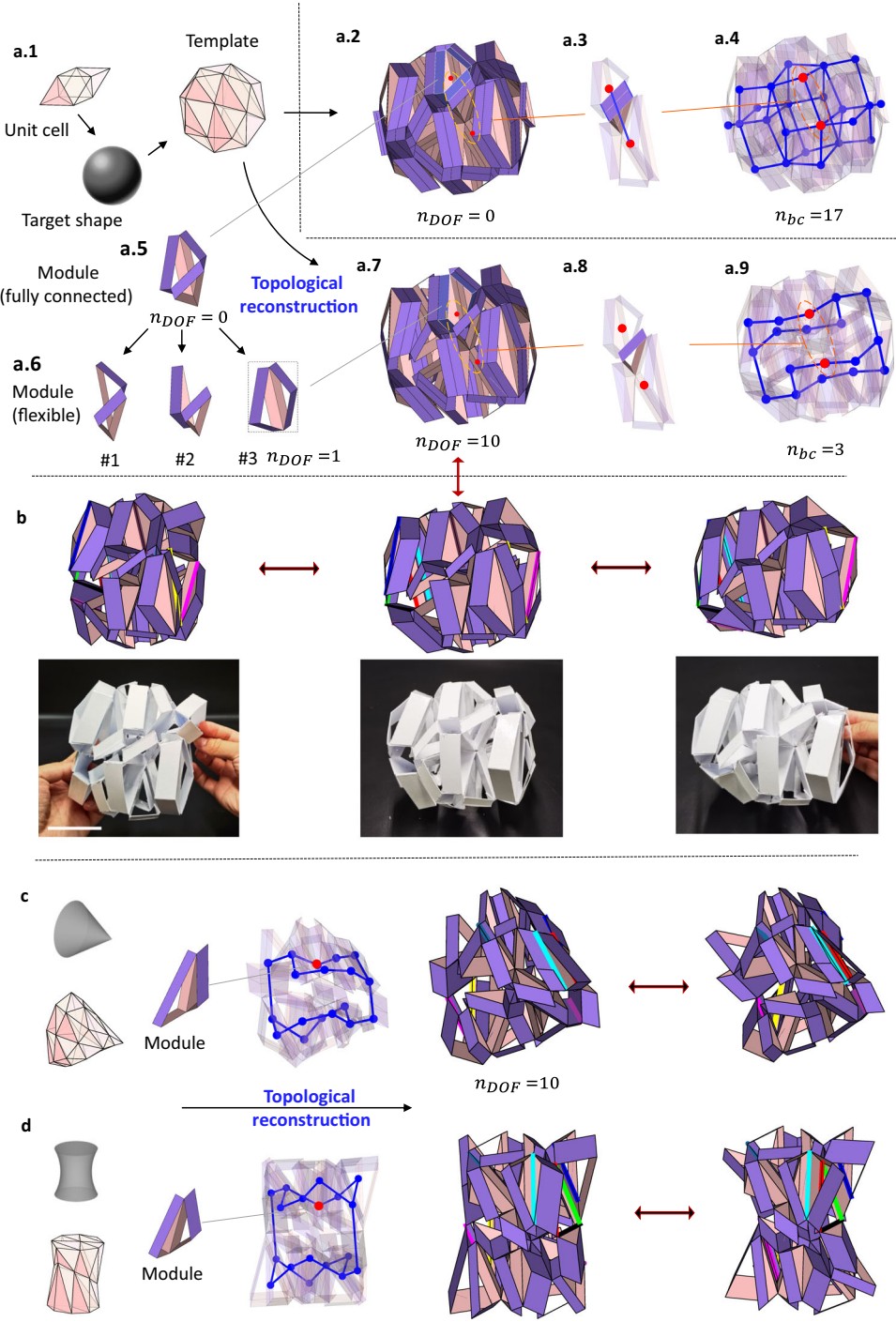

**Fig. 3 | Procedure to produce reconfigurability through topological reconstruction.** **a.**1 Sphere template constructed from the volumetric mapping ($2 \times 2 \times 2$) of a unit cell combining a tetrahedron and octahedron; **a.**2 Prismatic architected material after geometric modification; **a.**3 Two adjacent modules connected by prismatic tubes; **a.**4 Construction of a fully connected graph with basic cycles $n_{bc} = 17$; **a.**5 Rigid module fully connected with tubes; **a.**6 Modules with nonzero mobility; **a.**7 Deformable assembly with 10 DOFs after topological reconstruction; **a.**8 Two disconnected modules; and **a.**9 Reconstructed graph with fewer basic cycles, where $n_{bc} = 3$. **b** Transformation of a reconfigurable structure where the DOFs are denoted by 10 dihedral angles with colored hinges. The change of angles is discussed in Supplementary Movie 3 and Supplementary Note 4 of the SI. See details of the physical prototypes in Supplementary Note 4 of the SI. **c, d** Cone and hyperboloid templates having the same graph as **a.**9 with $n_{bc} = 3$, producing a reconfigurable structure with 10 DOFs. The size of the scale bar in b is 5 cm.

Figure 3c, d shows the topological reconstruction of modular origami structures generated by other templates — a cone and hyperboloid. When we select the same module topologically as the spherical structure in Fig. 3a.6 for the cone and hyperboloid structures in Fig. 3c, d, the modular origami structures produce the same spatial connection topology ($n_{bc} = 3$) and the same reconfigurable mobility (see Supplementary Movie 4). Therefore, the reconfigurability is controlled by the spatial connection of the modules and not by the geometry of the template.

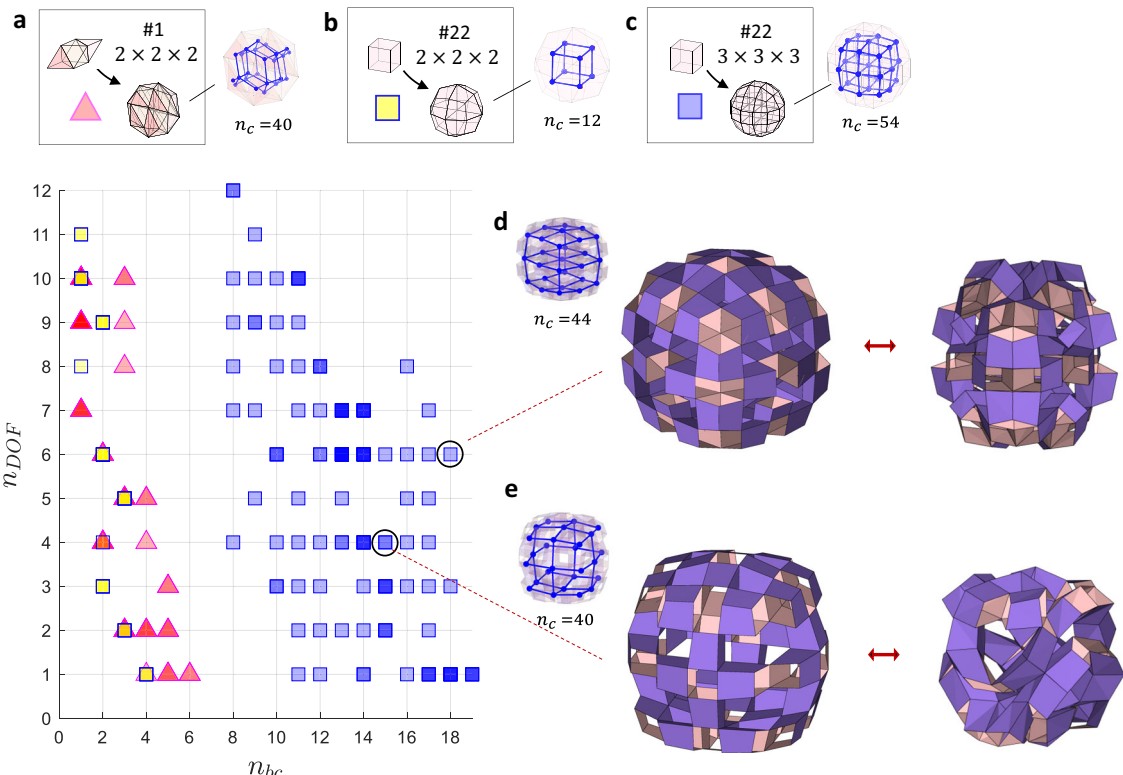

**Fig. 4 | Scatter plot on the inverse-designed mobility of 3D modular origami structures showing reconfigurable structures for various unit cells and mapping sizes; each point represents a combination of geometry and topology of modules.** **a** Template #1 was obtained using a $2 \times 2 \times 2$ mapping of a unit cell consisting of two tetrahedrons and one octahedron, **b** Template #22 was obtained using a $2 \times 2 \times 2$ mapping of a cubic unit cell; and **c** template #22 was obtained using a $3 \times 3 \times 3$ mapping of a cubic unit cell. **d** Selective sphere reconfigurable structures constructed using template #22 of (**c**) showing (**d**) uniaxial extension mode of a modular origami with $n_{bc} = 18$ and $n_{dof} = 6$, and (**e**) volumetric shrinking mode of a modular origami structure with $n_{bc} = 15$ and $n_{dof} = 4$. $n_c$ denotes the number of edges in the graph.

## Mobility evaluation

We implement transformation by obtaining the independent dihedral angles, as demonstrated in Figs. 3b–d and 4. Following the assumption of the rigid face and flexible hinges of prismatic modular origami, we identify the independent dihedral angles from a linearized constraint matrix. We apply additional kinematic constraints on every vertex: the distance between two vertices on an edge remains constant during reconfiguration (see Supplementary Fig. 8 in the SI for further details). While linearizing the constraints with a matrix form $\mathbf{J}_v$, we obtain $\mathbf{J}_v \cdot d\mathbf{v} = \mathbf{0}$[50], where $d\mathbf{v}$ is the infinitesimal displacement of vertices. We can also obtain the infinitesimal displacement of all the dihedral angles as $d\boldsymbol{\phi} = \mathbf{J}_h \cdot d\mathbf{v}$. In matrix form, the constraints can be expressed as

$$\begin{bmatrix} \mathbf{J}_v & \mathbf{0} \\ \mathbf{J}_h & -\mathbf{1} \end{bmatrix} \begin{bmatrix} d\mathbf{v} \\ d\boldsymbol{\phi} \end{bmatrix} = \mathbf{0}. \tag{11}$$

By calculating the reduced row echelon form of $\mathbf{J}\left( = \begin{bmatrix} \mathbf{J}_v & \mathbf{0} \\ \mathbf{J}_h & -\mathbf{1} \end{bmatrix} \right)$, we can determine the free variables in terms of $d\boldsymbol{\phi}$, which can produce movement of structural components, leading to reconfiguration (see Supplementary Note 3 for further details).

Figure 4 presents an example of reconfigurable structures assembled with varying geometry and topology of modules for a target DOF. Applying the geometric modification in Eqs. (3)–(6) and topological reconstruction in Eqs. (7)–(10) to 28 unit cells within a volumetric mapping size of 40[51,52], we obtain sphere reconfigurable structures for various target DOFs. Our algorithm provides multiple solutions, e.g., multiple combinations of geometry and topology for a target $n_{DOF}$. For example, unit cell #1 with a mapping size of $2 \times 2 \times 2$ and unit cell #22

with a mapping size of $3 \times 3 \times 3$ provide several topological options with varying $n_{bc}$ for target mobility, as shown in Fig. 4.

We plot the DOFs with varying connecting topology while demonstrating various deformation modes, including uniaxial extension and volumetric shrinkage, as shown in Fig. 4d, e (see the transformation in Supplementary Movie 5). The mobility can also be validated using a linear eigenmode analysis[16].

The same unit cell, e.g., a cubic unit cell, provides different reconfigurability depending on the variation of the topological connections, which is controlled by the mapping size. Small-sized polyhedrons with a $3 \times 3 \times 3$ mapping can provide a vaster design space in mobility than large-sized polyhedrons with a $2 \times 2 \times 2$ mapping because of the greater chance of available topology, as illustrated in Fig. 4. A unit cell consisting of two tetrahedrons and one octahedron with a $2 \times 2 \times 2$ mapping produces a maximum of 40 connections ($n_c = 40$), resulting in zero mobility, as illustrated in Fig. 3a. Even releasing the connection only produces a narrow range of design space in mobility because of the extruded tubular shapes of the tetrahedron and octahedron being a triangular lattice prism with kinematic mobility of zero.

## Discussion

3D curvilinear structures are ubiquitous in automotive, aerospace, and ocean engineering structures. To date, the design of architected materials has relied on the bottom-up design approach, using planar and orthogonal periodic tessellation of modules[23,53], which is convenient for structural design. However, this bottom-up approach has a critical limitation when constructing curvilinear 3D engineering and artistic structures, where the size and shape of the building blocks are no longer homogeneous in the design domain. The reconstruction of geometry and topology in this study provides a solution for designing

non-periodic and curvilinear 3D structures and their reconfigurability. The geometric reconstruction via volumetric mapping of polyhedrons followed by volumetric shrinkage can be used to construct a stress-free 3D curvilinear structure. The fully connected irregular modules produce spatial loops and constrain inter-and intra-modular motions, delivering a rigid yet stiff curvilinear structure. The topology reconstruction after geometric modification can provide solutions for an inverse design of reconfigurable structures with multi-DOF. The reconfigurable motion is controlled by the topology of the spatial connection of modules, not by the geometry of the templates. For advanced reconfigurability, one may need an optimization of shape transformation, which is beyond the scope of this work but can be explored near future.

The inverse design of 2D origami structures can generate a curvilinear surface with local control of the variable curvature[40,46]. Because of the instability (local snapping) of the out-of-plane deformation during reconfiguration, relatively high local energy is needed for the transformation[40]. The bottom-up approach with periodic modular origami structures can construct a curved shape by filling building blocks inside the curvilinear space[15,39]. However, the bottom-up approach cannot yield a smooth surface; it only provides discrete curves, whereas the geometric reconstruction in this work can produce a smooth 3D curvilinear structure. Notably, our inverse design of reconfigurability produces multiple solutions depending on the loop topology and geometry of modules.

Our top-down approach can expand the design space of modular origami to 3D non-periodic structures and 3D curvilinear geometries — spheres, hyperboloids, cones, twisted cylinders, and toruses, overcoming the limitation of planar and spatial tessellations for periodic structural design. Unlike 2D reconfigurable curvilinear origami structures, our 3D motion structures can be reconfigured into other shapes with multi-DOF and are frustration-free. This work will advance the design of 3D architected materials with reconfigurability and disrupt traditional periodic-tessellation-based 3D metamaterial design. The volumetric shrinkage of space-filled polyhedrons can be applied for both orthogonal and curvilinear coordinates, meaning that our inverse design method is universal and can be applied to both periodic and non-periodic structures. Our physical prototypes of 3D curvilinear structures fabricated using additive manufacturing validate the tunable motion, demonstrating untethered actuation by embedded hard magnets and moving external magnetic fields (see Supplementary Movies 3 and 4). The non-contact actuation of motion structures demonstrates the potential of active metamaterials, morphing devices, and soft robots on the mesoscale. Advanced micro-fabrication techniques such as two-photon lithography[7] and micro-stereolithography[11] can be applied to our reconfiguration systems on the microscale, e.g., for the design of battery electrodes[54] and microelectronic mechanical systems[55].

## Methods

### Volumetric mapping
To implement the volumetric mapping of a reference unit into the target shape, we use open-source software (GRAPHITE, ver. 3, developed by Bruno Lévy)[56], which conducts semi-discrete optimal transport[48]. For a given number of unit cells, we build a template. As the input to the software, we prepare two 3D object files: the boundary of a target shape and a unit cell with a given volumetric filling size, e.g., $3 \times 3 \times 3$, using a custom-made MATLAB script. We import these two 3D object files and volumetrically mesh them in the software while aligning the size and position of both meshes using a pre-processing tool in GRAPHITE. Next, we set the density of the sampled mesh to ~400,000, transporting the sampled mesh to fit the target shape. The software outputs a transported sample, which records the positions of these ~400,000 nodes in both the initial geometry and target shape. However, our template usually has fewer nodes; building the template

with a targeting profile requires matching the nodes of the unit cells with the sample in the initial shape, which is achieved using the *knnsearch* tool in MATLAB. We obtain the template by replacing the positions of the nodes of the unit cells with their counterparts in the transported sample with the target shape.

### Experiments
We prototype a rigid model using multi-material inkjet 3D printing (MultiJet, ProJet MJP 5600, 3D systems) and a MATLAB script to build an origami mesh model with a prescribed wall thickness. We fabricate reconfigurable prototypes with an assembly of paperboards (Silhouette Cameo) and additive manufacturing by stereolithography (SLA) by Form 3 (Formlabs). We print modular origamis and selectively embedded permanent magnets (NdFeB) to demonstrate a reconfigurable structure with remote control. We apply a rotational uniform magnetic field by a Halbach array composed of a circumferential array of permanent magnets. To validate our numerical simulation of reconfigurability with experiments, we fabricate a reconfigurable structure and use a 3D scanner (Einscan Pro) to capture the transformed shapes. We show the quantitative comparison of the independent dihedral angles of the reconfigurable structure between numerical simulation and 3D scanned measurement in Supplementary Note 4.

## Data availability
The data supporting this study's findings are included in the main text, Supplementary Information, or are available from the corresponding authors upon request.

## Code availability
The MATLAB model used to generate the modular origami structures is available via a GitHub repository link provided at the following URL: https://github.com/KaiXiao55/origami-architected-materials.git. Other algorithms necessary to reproduce the figures are available from the corresponding authors on request.

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

## Acknowledgements

J.J. acknowledges the support received from the Shanghai NSF (Award # 17ZR1414700), the National Natural Science Foundation of China (Award # 12272225), and the Research Incentive Program of Recruited Non-Chinese Foreign Faculty by Shanghai Jiao Tong University. We thank Y. Du for computational design of the prismatic materials with prescribed thickness. We also thank M. Huang, J. Sun, Y. Du, Y. Yu, and H. Wang for their assistance with the fabrication of the 3D-printed prototypes.

## Author contributions

K.X., X.Z., and J.J. proposed and designed the research project; X.Z. and J.J. supervised the project; K.X. and X.Z. designed the numerical

calculations; X.Z. and J.J. designed the physical models and experiments; K.X., Z.L., and B.Z. performed the numerical calculations, computational modeling, fabrication, and experiments; K.X. and X.Z. wrote the initial draft; J.J. revised the manuscript; all the authors reviewed the manuscript.

## Competing interests

The authors declare no competing interests.
