## [Peer Review File · Nature Communications]

Inverse design of 3D reconfigurable curvilinear modular origami structures using geometric and topological reconstructionsREVIEWER COMMENTS

Reviewer #1 (Remarks to the Author):

This article presents an inverse approach to designing reconfigurable origami-based 3D structures with curvilinear. The article is well-organized and the design approach may enrich the state-of-art in design method for the mechanical metamaterials community. Here are a few major and minor comments I would like to share with the authors for the revised version. Hope it may be helpful.

Major Comments:

- 1, Based on the literature, there are some inverse design principles in mechanical metamaterials in general. What are the uniqueness and advantages of the proposed approach compared with previous work? The author did mention parts of that in line 80. But it is not clear to me. The author may want to state them clearly.
- 2, It seems to me that the proposed approach highly relies on the reference templates as an initial condition for optimization. However, the author didn't clearly state how they choose the reference templates for each design (for example, fig.1 and fig.2).
- 3, Can the author demonstrate reconfigurable structures experimentally/physically based on the design principle (if my understanding is correct)?

Minor Comments:

- 1, In line 87, it stated "any extruding direction can be applied as long as there is no intersection of prismatic tubes". It is not clear to me.
- 2, similar problem, in line 122, it stated "To obtain 3D curvilinear structures with tunable motion, we developed a topology reconstruction method using a simple numerical algorithm. Adapting graph theory, we re-design the spatial loops of 3D curvilinear structures where neighboring modules are tightly connected." It is the key to the design but may need more explanation.
- 3, In fig. 3d, the experimental image on the right-hand side may need to be retaken to match the configuration shown in the schematics.
- 4, In line 175, it stated that "d_phi can produce the motion of the structures". If I'm correct, d_phi is the dihedral angle. The change of the dihedral angle may result in the movement of structural components leading to the motion.
- 5, in line 181, it stated that "The algorithm generated reconfigurability of a sphere motion structure....." What is a sphere motion structure?
- 6, few language usage issues in my opinion, for example, "notably" is observed throughout the article, line 240 "shaking up". The author may want to double-check during revision.

Reviewer #2 (Remarks to the Author):

The article by Xiao et al. describe an volumetric mapping approach for the design of 3D architected structures. The article is clearly written, and the goals set out in the beginning are clear. However, I have some major comments regarding the inverse design approach and the final concept for updating the

design of the structures that I list below (in no specific order):

1) While the authors aim is to implement an inverse design approach, I consider their approach more a mapping (as also mentioned in the title) than an inverse design approach. What I am missing in the approach of the authors is the optimisation of the structure towards some functionality, e.g. reconfigurability of the structure, or specific stiffness. At the moment, the authors map the structure onto a chosen geometry, but the functionality of the original structure is lost (i.e. foldability). What is specifically the objective function in their approach, and what are the constraints? Here, inverse design could have e.g. considered the mapping of the structure, given some constraints on the available folding degrees of freedom.

2) I also want to argue that in their approach the initial chosen design of the structure is important to be able to achieve a successful mapping. For example, if the final shape has a hole (e.g. the torus), the initial design also requires a hole. Therefore, still considerable knowledge is required to use their approach, which limits the generality.

3) The article is builds upon previous work (Reference [16]). In that article, the authors propose to use the reconfigurable metamaterials directly as a material and effectively shape (i.e. machine) the metamaterials in certain geometries just like you would do with any other material. From that perspective, I wonder why the mapping is needed at all, since as also pointed out by the authors the mapping causes the structures to lose their degrees of freedom, which in turn is then solved by a heuristic approach to remove connected to introduce foldability. Can the authors comment on the benefit of their approach in relation to the original approach?

4) Related to (3), the images and numerical analysis is (exactly) the same as previous work [16], but no proper reference is given when introducing e.g. the numerical analysis.

5) Regarding the experiments, I was impressed by the structures they achieved and the actuation of these structures using magnets. However, the experiments are only qualitative and no comparison is made to the simulations. In fact, I found the details on experiments and the discussion very limited, while I found the achievements considerable compared to the method introduced for the inverse design. It would be great if the authors can include more details and results of the experiments.

6) Looking at Supplemental Figure 5, the mapping is shown for different unit number (for the same geometry). Is there an effect of the unit size on the mapping and on the reconfigurability of the structure? What is the effect of unit size on their approach to remove faces to reintroduce foldability?

7) Finally, for the motion achieved after removing faces are limited to what seems to be hinging motions of two rigid parts connected by two faces. Originally the templates produced microscopic folding modes where the folding behaviour was mostly throughout the whole structure. Can the authors comments on this?

Reviewer #3 (Remarks to the Author):

Kai Xiao et al. present an approach for discretization and reconstruction of 3D objects using prismatic origami units and then modifying them into flexible origami objects by reducing the number of connections. In general, their work is a rational way of creating complex transformable objects and might serve as a method to design flexible metamaterials. I believe this work is suitable for publication in Nature Communications. The authors should then consider the comments below. Although the text has been written in good English, I advise improving the article's readability by better highlighting the points such as predisposing for flexibility and evaluating the degrees of freedom, and by avoiding moving forward and backward between the main text and the supplementary information while possible.

Comments:

The last 3 sentences in the abstract look to convey a similar message. I should rewrite these sentences.

Can you explain how you choose the reference space-filling and how critical is your choice for mobility?

From the text, it can be understood that the deformed polyhedrons shrink uniformly with respect to their centroids. However, the way it has been explained in line 73 "Now, we spatially shrink the deformed polyhedrons ..." makes it hard to understand without moving forward and backward between the text and figure 1d. Authors then may like to rewrite this part.

The example of 2D transformation is trivial, and it is better to put it in ESI.

A better rearrangement of the panels in figure 1 will help to better follow the process of transformation into a modular origami structure.

In line 120, the authors note that irregular polyhedron units are analogous to linkage-bar mechanisms. However, it has not been illustrated how they are equivalent to each other.

The section "Construction of tunable motion" describes the predisposing geometry for flexibility and evaluation the mobility. It might be better to divide this section into two separate sections.

It seems that the reconstructed geometries have the freedom to shear on a specific plane rather than exhibiting global reconfigurability. Is that correct?

Does your approach depend on system size? Can you evaluate that on one of the examples from figure S5, for example?

Can you compare your approach with a bottom-up approach in which the DoF has been defined based

on the kinematics of linkage bar mechanisms, and then the rigid links have been replaced with shrank units?

Reviewer #1

This article presents an inverse approach to designing reconfigurable origami-based 3D structures with curvilinear. The article is well-organized, and the design approach may enrich the state-of-the-art design method for the mechanical metamaterials' community. Here are a few major and minor comments I would like to share with the authors for the revised version. Hope it may be helpful.

Major Comments:

1, Based on the literature, there are some **inverse design principles** in mechanical metamaterials in general. What are the **uniqueness** and **advantages** of the proposed approach compared with previous work? The author did mention parts of that in line 80. But it is not clear to me. The author may want to state them clearly.

We modified the fourth paragraph of the introduction by emphasizing the **uniqueness** and advantage of our proposed approach compared with previous work.

"The top-down approach to the design, often called 'inverse design,' of mechanical metamaterials has been explored to find tessellated microstructures for targeting physical properties such as anisotropic stiffness. Topology optimization has been a typical approach to finding microstructures³⁶⁻³⁸. Very few studies have explored the inverse design of 3D architected structures³⁹. The existing inverse-design methods can only be applied to 2D curvilinear surfaces with origami and kirigami⁴⁰⁻⁴⁷ and not to volumetric 3D spatial curvilinear geometries and their reconfigurability. The current work explores an inverse design of 3D reconfigurable architected origami materials. Without tessellating a constant building block, our method produces volumetric gradient cells mapped into complex curvilinear 3D geometries, followed by topological reconstruction of modules."

Line 80 in the initial manuscript was **not** directly related to the inverse design but the last step to generate prismatic architected materials. Without a centroid connection, we cannot generate modules in the templates for curvilinear geometries. The separation method in reference [16] only applies to the regular polyhedrons with flat surfaces and cannot map into curvilinear 3D geometries. Our method of bridging centroids of irregular polyhedrons can map into any free-form surface, as indicated in Figure 1d of the revision. Moreover, the bridging centroids in this work perform superior for templates with concave and convex polyhedrons, where concave polyhedron's surface normal vectors cross each other if [16] is used.

2, It seems to me that the proposed approach highly relies on the reference templates as an initial condition for optimization. However, the author did not clearly state how they choose the reference templates for each design (for example, fig.1 and fig.2).

In the revision, there are some changes to the terminologies. We removed the reference templates. Instead, we use **unit cells**, as shown in Figure 1a of the revision. Also, we changed the transformed templates to **templates**. The selection of a **unit cell's** geometry does not affect the geometric reconstruction; one can generate the geometric reconstruction (volumetric mapping and shrinkage) with any unit cell regardless of a unit cell's geometry, as shown in Figure 1 and Figure S3. The initial selection of geometry affects reconfigurability at a certain level; e.g., tetrahedrons and octahedrons produce an extruded triangular lattice prism with zero kinematic mobility. We discussed this finding before Figure 4 of the revision:

“A unit cell consisting of two tetrahedrons and one octahedron with a $2 \times 2 \times 2$ mapping produces a maximum of 40 connections ($n_c = 40$), resulting in zero mobility, as shown in Figure 3a. Even releasing the connection only produces a narrow range of design space in mobility, due to the extruded tubular shapes of tetrahedron and octahedron being a triangular lattice prism whose kinematic mobility is zero”.

3, Can the author demonstrate reconfigurable structures experimentally/physically based on the design principle (if my understanding is correct)?

We carefully redesigned our algorithm and demonstrated our design principle with physical prototypes in the revision. Figure 3b, Supplementary Videos 3-5, and Section 4.4 of Supplementary Information in the revision demonstrate the reconfigurable structures constructed by the design principle in Figures 2 and 3.

Minor Comments:

1, In line 87, it stated “any extruding direction can be applied as long as there is no intersection of prismatic tubes”. It is not clear to me.

After the revision, we rewrote the sentences to clearly explain the construction of tubes on the boundary surface of exterior shrunk polyhedrons after Equation (2):

“We apply a different construction method for the tube on the exterior boundary surface of a shrunk polyhedron before the volumetric shrinkage. The tubular length, d_b , generated along the exterior boundary surface in Figure 1d is determined by the distance between the template and the exterior surface of a shrunk polyhedron along its normal direction”.

2, Similar problem, in line 122, it stated “To obtain 3D curvilinear structures with tunable motion, we developed a topology reconstruction method using a simple numerical algorithm. Adapting graph theory, we redesign the spatial loops of 3D curvilinear structures where neighboring modules are tightly connected.” It is the key to the design but may need **more explanation**.

We removed the sentence in the revision. We revised our algorithm of reconfigurability using a geometric modification (Figure 2) and topological reconstruction (Figure 3) in the revision. Equations (3) – (10) in the revision clearly describe the algorithm.

3, In fig. 3d, the experimental image on the right-hand side may need to be retaken to match the configuration shown in the schematics.

In the revision, we have changed the prototype figures, as shown in Figure 3b, which better matches our algorithm because we proceed with geometric modification and topological reconstruction (see Figures 2 and 3).

4, In line 175, it stated that “ d_ϕ can produce the motion of the structures.” If I’m correct, d_ϕ is the dihedral angle. The change of the dihedral angle may result in the movement of structural components leading to the motion.

Yes, you are right. We revised the sentence as you suggested:

"...we can determine the free variables in terms of $d\phi$, which can produce movement of structural components, leading to reconfiguration".

5, In line 181, it stated that "The algorithm generated reconfigurability of a sphere motion structure...."
What is a sphere motion structure?

Initially, we meant a motion structure with a sphere template. However, we removed the sentence in the revision to avoid confusion.

6, few language usage issues in my opinion, for example, "notably" is observed throughout the article, line 240 "shaking up." The author may want to double-check during revision.

We avoid overusing "notably" in the revision. We also removed "shaking up" in the revision. In the revision, we paid extra care in selecting professional words with additional professional editing services.

Reviewer #2

The article by Xiao et al. describes a volumetric mapping approach for the design of 3D architected structures. The article is clearly written, and the goals set out, in the beginning, are clear. However, I have some major comments regarding **the inverse design approach** and **the final concept** for updating the design of the structures that I list below (in no specific order):

1) While the authors' aim is to implement an inverse design approach, I consider their approach more a mapping (as also mentioned in the title) than an inverse design approach. What I am missing in the approach of the authors is **the optimisation of the structure towards some functionality**, e.g., reconfigurability of the structure, or specific stiffness. At the moment, the authors map the structure onto a chosen geometry, but the functionality of the original structure is lost (i.e., foldability). What is specifically the **objective** function in their approach, and what are the **constraints**? Here, an inverse design could have, e.g., considered the mapping of the structure, given some **constraints** on the available folding degrees of freedom.

During the revision period, we reconstructed our inverse design algorithm on the reconfigurability of 3D curvilinear modular origami structures – geometric reconstruction and topological reconstruction (with geometric modification). We redesigned the manuscript to show that our objective is not just mapping but a systematic design approach to reconfigurability. The main processes for reconfigurability in the revision consist of two steps - **geometric reconstruction** and **design of reconfigurability** (geometric modification and topological reconstruction).

Geometric reconstruction is a volumetric mapping process of unit cells into a 3D curvilinear shape followed by volumetric shrinkage, shown in Figure 1 of the revision. Actual inverse design for functionality – reconfigurability starts from the geometric modification and topological reconstruction, shown in Figures 2 and 3, respectively, in the revision. The **geometric modification** in Equations (3)–(6) is the postprocess of the geometric reconstruction, and the preprocess of the topological reconstruction, which releases the kinematic constraints generated by the previous curvilinear mapping and facilitates the construction of foldable modules – an essential procedure for the topological reconstruction. The **topological reconstruction** is a core part of the inverse design of reconfigurability. The algorithm in Equations (7) – (10) searches for mobility and the corresponding topology and geometry for sets of modules. As the reviewer suggested, we set an objective function with constraints for our optimization problems in the revision.

2) I also want to argue that in their approach the initial chosen design of the structure is important to be able to achieve a successful mapping. For example, if the final shape has a hole (e.g., the torus), the initial design also requires a hole. Therefore, still considerable knowledge is required to use their approach, which limits the generality.

Thank you for bringing up a particular case. You are right; if we use the reference template in the initial manuscript, we may require an initial guess to select the reference template, limiting the generality. To resolve this argument, we modified the mapping procedure by removing the construction process of the reference templates. Instead, we directly map **unit cells** to any prescribed shapes, whether it is a torus or not, as shown in Figure 1 of the revision. Therefore, our new approach does not require considerable knowledge of the selection of unit cells.

3) The article is built upon previous work (Reference [16]). In that article, the authors propose to use the reconfigurable metamaterials directly as a material and effectively shape (i.e., machine) the metamaterials in certain geometries just like you would do with any other material. From that perspective, I wonder why the mapping is needed at all, since as also pointed out by the authors the mapping causes the structures to lose their degrees of freedom, which in turn is then solved by a heuristic approach to remove connected to introduce foldability. Can the authors comment on the **benefit** of their approach in relation to the original approach?

The original work [16] was an excellent pioneering study to open 3D reconfigurable modular origami structures but was limited to the reconfigurable design in a periodic tessellation of modules. Unfortunately, practical engineering and artistic structures, whose shapes are mostly 3D curvilinear (e.g., automotive and aerospace structures with various curvatures), require the size and shape of the building blocks to no longer be homogeneous in the design domain. Our method fills arbitrary unit cells to 3D curved volumes with spatially gradient tiling through a volumetric mapping. The volumetric mapping for 3D curvilinear shapes is necessary because filling unit cells with periodic tessellation in 3D curvilinear shapes produces discretization (defect) on the boundary of the assembly.

Unlike the previous work (Ref. [16]) with periodic structures, modular assemblies' reconfigurability in this study does not precisely follow the unit cell's motion due to the nonperiodic tessellation of spatially gradient cells, which incredibly challenges us in implementing mobility. In the revision, we build an algorithm for obtaining target mobility via geometric modification and topological reconstruction, providing global reconfigurability.

Note that we also improved the quality of reconfigurability with the advanced algorithm – geometric modification and topological reconstruction in the revision. Our initial manuscript only showed a local deformation by a heuristic method, but the revision demonstrates the global deformation by robust optimization logic.

4) Related to (3), the images and numerical analysis is (exactly) the same as previous work [16], but no proper reference is given when introducing, e.g., the numerical analysis.

We gave credentials to the previous frontier work [16] on the synthesis of 3D modular origami by citing several times the design principle and numerical analysis of [16] in the revision, including the second paragraph in the introduction, the first paragraph of Section II – Synthesis of nonperiodic modular origami, and the paragraphs after Equations (1) and (11).

5) Regarding the experiments, I was impressed by the structures they achieved and the actuation of these structures using magnets. However, the experiments are only qualitative and no comparison is made to the simulations. In fact, I found the details on experiments and **the discussion very limited**, while I found the achievements considerable compared to the method introduced for the inverse design. It would be great if the authors can include more details and results of the experiments.

We demonstrated our design principle with physical prototypes based on our reconstructed algorithm in the revision. Figure 3b, Supplementary Videos 3-5, and Figures S10-S12 in the Supplementary Information demonstrate reconfigurable structures constructed by the design principle in Figures 2 and 3.

As suggested by the reviewer, we added the fabrication method of reconfigurable structure with magnetic control in Section 4.3 of the Supplementary Information. Moreover, we quantitatively checked the reconfigurability by comparing 3D scanned data with our simulation in Section 4.4 of the Supplementary Information.

6) Looking at Supplemental Figure 5, the mapping is shown for different unit numbers (for the same geometry). Is there an effect of the unit size on the mapping and on the reconfigurability of the structure? What is the effect of unit size on their approach **to remove faces to reintroduce foldability?**

A denser mapping requires a severe volumetric change of the reference unit during the mapping process, as shown in Section 1.3 in the Supplementary Information of the revision. A small-sized unit cell with a $3 \times 3 \times 3$ mapping can provide more vast design space in mobility than a large-sized unit with a $2 \times 2 \times 2$ mapping due to the greater chance of available topology, as shown in Figure 4. (We mentioned the size effect before Figure 4 in the revision.)

Also, note that we do not simply remove faces in the revision; we search through all possible combinations of geometry and topology of modules to obtain mobility of assemblies.

7) Finally, for the motion achieved after removing faces are limited to what seems to be hinging motions of two rigid parts connected by two faces. Originally the templates produced microscopic folding modes where the folding behaviour was mostly throughout the whole structure. Can the authors comment on this?

During the geometric reconstruction – volumetric mapping of unit cells and volumetric shrinkage of irregular polyhedrons, the volumetrically foldable functionality of modules is destroyed, which is inevitable with 3D curvilinear modular origami structures due to the nonhomogeneous distribution of deformed polyhedrons.

Therefore, we resolve the initial immobility issue in the revision by adding the geometric modification algorithm in Figure 2 and Equations (3)-(6). After the geometric modification, we also revised our algorithm of topological reconstruction in Equations (7)-(10) and Figure 3 by searching for all possible reconfigurability covering i) all possible geometric and topological options of modules and ii) the connection of adjacent modules. The revised algorithm provides better global reconfigurability, as shown in Figures 3 and 4 and Supplementary Videos 3-5.

Reviewer #3

Kai Xiao et al. present an approach for discretization and reconstruction of 3D objects using prismatic origami units and then modifying them into flexible origami objects by reducing the number of connections. In general, their work is a rational way of creating complex transformable objects and might serve as a method to design flexible metamaterials. I believe this work is suitable for publication in Nature Communications. The authors should then consider the comments below. Although the text has been written in good English, I advise improving the article's readability by better highlighting the points, such as **predisposing for flexibility** and **evaluating the degrees of freedom**, and by avoiding moving forward and backward between the main text and the supplementary information while possible.

Comments:

1. The last 3 sentences in the abstract look to convey a similar message. You should rewrite these sentences.

We deleted the last-second sentence in the revision and rewrote the first and third sentences in the abstract.

2. Can you explain how you choose **the reference space-filling** and how critical your choice is for mobility?

We choose any combination of polyhedrons for a unit cell in the revision; we do not impose any preference to select specific polyhedrons for a generalized purpose of space-filling. The mobility control starts after the geometric reconstruction – the construction of templates by volumetric mapping and volumetric shrinkage of deformed polyhedrons, as shown in Figures 1a-1e in the revision.

The spatial connection of modules primarily determines the mobility of reconfigurable structures, as demonstrated in Figure 4. However, a module, which is a tubular origami by tubular extrusion on the surface of deformed polyhedrons, also contributes to mobility. For example, a unit cell composed of two tetrahedrons and one octahedron with a $2 \times 2 \times 2$ mapping produces a maximum of 40 connections ($n_c = 40$), resulting in zero mobility, as shown in Figure 3a. Even releasing the connection only produces a narrow range of design space in mobility due to the extruded tubular shapes of tetrahedron and octahedron being a triangular lattice prism whose kinematic mobility is zero. (We added this discussion before Figure 4 in the revision.)

We also discussed the effect of mapping density on reconfigurability before Figure 4 and Section 1.3 of the Supplementary Information in the revision.

3. From the text, it can be understood that the deformed polyhedrons shrink uniformly with respect to their centroids. However, the way it has been explained in line 73 "Now, we spatially shrink the deformed polyhedrons ..." makes it hard to understand without moving forward and backward between the text and figure 1d. Authors then may like to rewrite this part.

We rewrote this part by clearly separating the volumetric mapping and volumetric shrinkage of the geometric reconstruction after Equation (2) in the revision.

4. The example of 2D transformation is trivial, and it is better to put it in ESI.

We moved Figure 1g of the initial manuscript to Figure S1 in the Supplementary Information.

5. A better rearrangement of the panels in figure 1 will help to better follow the process of transformation into a modular origami structure.

We rearranged Figure 1 in the revision to clearly describe the geometric reconstruction – volumetric mapping and volumetric shrinkage.

6. In line 120, the authors note that irregular polyhedron units are analogous to linkage-bar mechanisms. However, it has not been illustrated how they are equivalent to each other.

Initially, we thought the planar extruded tubes from the irregular polyhedron were similar to a linkage mechanism. However, we brought the geometric modification of Equations (3)-(6) to the revision, which provides spatial foldability. The added foldability constraints make us challenging to describe the analogy with planar or simple spatial mechanisms. Therefore, we deleted the previous sentence in the revision. However, we added the foldability of cross-sectional tubular geometries, which are analogous to a 2D mechanism in Figure S5 of the Supplementary Information in the revision.

7. The section "Construction of tunable motion" describes the predisposing geometry for flexibility and evaluation of mobility. It might be better to divide this section into two separate sections.

As suggested, we divided the section into two in the revision.

8. It seems that the reconstructed geometries have the freedom to shear on a specific plane rather than exhibiting global reconfigurability. Is that correct?

The volumetric foldability of modules is destroyed during the geometric reconstruction - volumetric mapping of polyhedrons and volumetric shrinkage of deformed polyhedrons. Therefore, we resolve the immobility issue in the revision by adding the geometric modification algorithm in Figure 2 with Equations (3)-(6). After the geometric modification, we also revised our algorithm of topological reconstruction in Equations (7)-(11) and Figure 3 by searching for all possible reconfigurability covering i) all possible geometric and topological options of modules and ii) the connection of adjacent modules. The revised algorithm provides global reconfigurability, as shown in Figures 3 and 4 and Supplementary Videos 3-5.

9. Does your approach depend on **system size**? Can you evaluate that on one of the examples from figure S5, for example?

We tested the effect of system size on reconfigurability, showing the results in Figure 4. A small-sized unit with a $3 \times 3 \times 3$ mapping (a large system size) can provide a vaster design space in mobility than a large-sized unit with a $2 \times 2 \times 2$ mapping (a small system size) due to the greater chance of available topology with an extensive system size. We also investigated the size effect on mapping in Section 1.3 of the Supplementary Information.

10. Can you compare your approach with a bottom-up approach in which the DoF has been defined based on **the kinematics of linkage bar mechanisms**, and then the rigid links have been replaced with shrank units?

We added the comparison between bottom-up and top-down approaches in Section 2.3 of the Supplementary Information. As shown in Figure S7, our algorithm validates the bottom-up approach and provides more geometric and topological options for a target DOF.

REVIEWERS' COMMENTS

Reviewer #1 (Remarks to the Author):

Dear Authors, thank you for your efforts in incorporating my comments. The revised version makes the argument clear and convincing.

Reviewer #2 (Remarks to the Author):

I am happy with the changes made by the authors, yet I find the scope of the article and the focus on experimental validations still limited as also mentioned in my first comments. From a technical perspective, the authors have made considerable changes with respect to the inverse design approach, and given that they use genetic algorithms to solve the objective function, it is now not only a heuristic approach, but also an inverse approach. I do find the objective function a bit limited, in that they optimize for degrees of freedom, rather than e.g. optimize for shape transformation. I leave it up to the authors if they would like to also comment about that in e.g. their conclusion, or that they already have evidence that this could work.

Reviewer #3 (Remarks to the Author):

The authors accordingly addressed all the comments, and the manuscript is suitable for publication in its current form.

Reviewer #2

I am happy with the changes made by the authors, yet I find the scope of the article and the focus on experimental validations still limited as also mentioned in my first comments. From a technical perspective, the authors have made considerable changes with respect to the inverse design approach, and given that they use genetic algorithms to solve the objective function, it is now not only a heuristic approach but also an inverse approach. I do find the objective function a bit limited, in that they optimize for degrees of freedom, rather than e.g. optimize for shape transformation. I leave it up to the authors if they would like to also comment about that in e.g. their conclusion, or that they already have evidence that this could work.

We added our method's limitation to the discussion in the revision:

"For advanced reconfigurability, one may need an optimization of shape transformation, which is beyond the scope of this work but can be explored near future".